# Analysis of the Potential Range of Mountain Pine-Broadleaf Ecotone Forests and Its Changes under Moderate and Strong Climate Change in the 21st Century

**DOI:** 10.3390/plants12213698

**Published:** 2023-10-26

**Authors:** Nikolay Fedorov, Svetlana Zhigunova, Pavel Shirokikh, Elvira Baisheva, Vasiliy Martynenko

**Affiliations:** Ufa Institute of Biology, UFRC RAS, Ufa 450054, Russia; zigusvet@yandex.ru (S.Z.); elvbai@mail.ru (E.B.); vb-mart@mail.ru (V.M.)

**Keywords:** relict pine-broadleaf ecotone forests, MaxEnt, climate change, habitat suitability

## Abstract

Climatic changes have a significant impact on the composition and distribution of forests, especially on ecotone ones. In the Southern Ural, pine-broadleaf ecotone forests were widespread during the early Holocene time, but now have persisted as relic plant communities. This study aimed to analyze the current potential range and to model changes in habitat suitability of relic pine-broadleaf ecotone forests of the suballiance *Tilio-Pinenion* under scenarios of moderate (RCP4.5) and strong (RCP8.5) climate change. For modelling, we used MaxEnt software with the predictors being climate variables from CHELSA Bioclim, the global digital soil mapping system SoilGrids and the digital elevation model. In the Southern and Middle Urals, climate change is expected to increase the areas with suitable habitat conditions of these forests by the middle of the 21st century and decrease them in the second half of the century. By the middle of the 21st century, the eastern range boundary of these forests will shift eastward due to the penetration of broad-leaved tree species into coniferous forests of the Southern Ural. In the second half of the century, on the contrary, it is expected that climate aridization will again shift the potential range border of these forests to the west due to their gradual replacement by hemiboreal coniferous forests. The relationship between the floristic composition of pine-broadleaf forests and habitat suitability was identified. In low and medium habitat suitability, pine-broadleaf forests contain more nemoral species characteristic of deciduous forests of the temperate zone, and can be replaced by broadleaf forests after thinning and removal of pine. In the Volga Upland, suitable habitats are occupied by pine-broadleaf forests of the vicariant suballiance *Querco robori-Tilienion cordatae*. Projected climatic changes will have a significant impact on these ecotone forests, which remained completely unaltered for a long time.

## 1. Introduction

The Southern Ural (SU)—the southern and widest part of the Ural Mountains system—is the transitional zone between the three largest forest biomes of Eurasia: the mesic deciduous and mixed forests of temperate Europe, hemiboreal pine and birch-pine herb-rich open forests of Siberian type, and holarctic coniferous boreal taiga forests. The eastern boundary of the oak distribution (*Quercus robur* L.), maple (*Acer platanoides* L.) and elm passes along the Ural ridge. Deciduous linden-oak-maple forests with the presence of elm (*Ulmus glabra* Huds.) dominate on the western slope of the SU [1]. In the central part of the SU, deciduous forests are replaced mainly by pine-larch coniferous forests, which in the Pleistocene were distributed from Western Europe to the eastern foothills of the SU [2]. The role of pine (*Pinus sylvestris* L.) in the vegetation cover of the SU changed according to significant climate fluctuations in the Pleistocene and Holocene, as well due to the expansion of deciduous species into coniferous forests [3]. In the transitional zone between pine-larch coniferous and deciduous forests of the SU, mixed pine-broadleaf ecotone forests have long existed, characterized by the predominance of linden (*Tilia cordata* Mill.), and the presence of oak (*Quercus robur*), maple (*Acer platanoides*) and herb species typical for nemoral deciduous forests of a temperate zone and hemiboreal forests of the Siberian type. In accordance with floristic classification, these ecotone forests are included into class Carpino–Fagetea sylvaticae Jakucs ex Passarge 1968, order Carpinetalia betuli P.Fukarek 1968, alliance Aconito lycoctoni–Tilion cordatae Solomeshch et Grigoriev in Willner et al. 2016, suballiance Tilio cordatae-Pinenion sylvestris Shirokikh et al. 2021 [4]. Explanations: This sentence lists syntaxon names according to the code of Phytosociological Nomenclature (https://www.researchgate.net/publication/339794868_International_Code_of_Phytosociological_Nomenclature_4th_edition, accessed on 23 October 2023). Literature references are not required in this context. Simulations of the distribution of vegetation types in Europe during the Last Glacial Maximum using MaxEnt, based on modern data from the Siberian region, which has a climate similar to the European glacial climate, revealed the presence of areas with high and moderate climate suitability for temperate light-coniferous forests in Southern and Eastern Europe [5]. The forests with *Pinus sylvestris* were rapidly replaced by forests with *Picea abies* (L.) H. Karst. and temperate deciduous forests about 10,500 years ago [3]. Thus, modern pine-broadleaf forests of the SU represent relict ecotone forests similar to forests that were widespread in Southern and Eastern Europe during the late Pleistocene and the early Holocene time [2,6].

Changes in forest health under the influence of global climate change have been recorded worldwide [7,8,9,10,11,12,13,14,15,16,17]. In many temperate regions of Europe, growing season temperature has increased by more than 1 °C over the last four decades, resulting in a higher atmospheric evaporative demand [18,19]. Global climate change is affecting the distribution of different types of forest communities. Widespread declines in tree viability and increases in tree mortality in temperate forests and elsewhere have been linked to increasing aridification of the climate in recent times [20,21,22,23,24,25]. In particular, at the low-elevation and low-latitude range of the temperate tree species range, their growth is increasingly restricted by summer drought and heat waves [26,27,28,29,30]. But, high temperatures and a longer growing season can also have a positive effect, leading to increased tree growth and productivity, as well as new colonization of areas that were previously unoccupied by trees at high elevations [31,32]. Climate change affects the growth rate and phenological development of trees [25,33,34,35,36], leading to a shift to earlier growing season start dates, increasing the risk of frost damage to budding leaves in the temperate zone [37,38,39,40]. The combination of these factors in some cases leads to a change in the ratio of tree species and a change in the dominant tree species. For example, in mountain spruce-beech forests in the Czech Republic, there is an increase in the proportion of beech [41]. These trends may increase with climate change. Similar processes occur in the SU. In the SU over the past 100 years, the average annual temperature has increased by 1.4–1.5 °C [42]. The Southern Ural region is currently experiencing changes in the distribution boundaries of tree species, as well as plant community boundaries [12,43,44], including a shift of the upper boundaries of the mountain forests and a reduction in the distribution of mountain tundra communities [45], as well as a shift in the border ranges of broad-leaved tree species [46].

The tree species ranges are thought to be changing at a slower rate than climate change [47]. Periods of relaxation—the equilibrium state of ecotone vegetation—can last hundreds of years in some cases [48]. However, their condition can be dramatically disturbed when climate change accelerates, especially when combined with anthropogenic impacts. In this case, with climate warming, shade-tolerant undergrowth of some broad-leaved species may emerge into the tree layer [41,49]. Thus, climate change can contribute both to expansion of the ranges of ecotone communities and the formation of pure stands of species better adapted to new climatic conditions in some habitats, which are currently occupied by ecotone forest communities.

This study aimed to analyze the current potential range and to model changes in the habitat suitability of relic pine-broadleaf ecotone forests of the suballiance *Tilio-Pinenion* under scenarios of moderate (RCP4.5) and strong (RCP8.5) climate change.

## 2. Results

### 2.1. Current Potential Range 

The model of the current potential range of pine-broadleaf forests of the suballiance *Tilio-Pinenion* with different habitat suitability was calculated based on geo-referenced points (localities of geobotanical relevés) of these forests, established in the area of its real distribution in the mountain-forest zone of the SU and on the Ufa plateau (Figure 1). The current potential range reflects the distribution of habitats with conditions which are suitable for these forests at present. The model of the current potential range of pine-broadleaf forests has an AUC of 0.99, which corresponds to a high quality of the model [50]. The threshold value of habitat suitability is 0.23. The potential range map shows habitats of low suitability (0.24–0.49), medium suitability (0.50–0.75) and high suitability (0.75–1.00). Highly suitable habitats are concentrated in the SU and on the Ufa Plateau (Republic of Bashkortostan).

Also, suitable habitats were identified outside the area of main distribution of pine-broadleaf forests of the suballiance *Tilio-Pinenion*. In the Volga Upland in the Ulyanovsk and Saratov regions, small areas with low and medium suitability of habitat conditions are identified. There are only small areas with low suitability of habitat conditions in the Bugulma-Belebey Upland, Republic of Mordovia and Penza Oblast. Besides the SU, the highly suitability habitats are found in small areas near the border of two geomorphologic regions of the Volga Upland and Volga Lowland, i.e., in the Zhigulevsky Nature Reserve and the Samarskaya Luka National Park (Samara region) (Figure 1).

### 2.2. Features of Floristic Composition of Pine-Broadleaf Forests of the Tilio-Pinenion Suballiance Depending on the Degree of the Habitat Suitability

Among 178 georeferenced localities of pine-broadleaf forests of the suballiance *Tilio-Pinenion*, 14 are confined to low, 74 to medium and 90 to high habitat suitability (Figure 2). The largest areas of habitats with high suitability are concentrated in the southern part of the current range of pine-broadleaf forests in the SU. In the Ufa Plateau area, most habitats have medium habitat suitability. There are significant differences in the structure of tree and herb layers in the communities of the habitats with different degrees of suitability.

Figure 3 shows differences in the projective cover of broad-leaved species (*Acer platanoides*, *Tilia cordata*, *Quercus robur*, *Ulmus glabra*) in pine-broadleaf forests located in the areas with different habitat suitability. In the communities within areas with low habitat suitability, the projective cover of broad-leaved trees (both in the general tree layer and undergrowth) is estimated to a total of 20%, and it is much higher than in the forests located in the sites with medium and high habitat suitability (this trend has been observed for all broad-leaved trees, except *Quercus robur*). In terms of floristic composition, pine-broadleaf forests located in areas with low habitat suitability are close to deciduous forests of the *Aconito-Tilenion* suballiance. As the habitat suitability increases, the projective cover of broad-leaved species among large undergrowth has reduced. The communities located in the sites with high habitat suitability are characterized by weak shading and high cover of small undergrowth formed by linden, which is rarely presented in the second and first tree layers of these forests.

Also, the significant differences were revealed in the composition of the herb layer of pine-broadleaf forests growing in habitats with different suitability (Figure 4). The proportion of species typical for hemiboreal forests of the class *Brachypodio-Betuletea* (*Angelica sylvestris* L., *Brachypodium pinnatum* (L.) Beauv., *Bupleurum longifolium* L., *Calamagrostis arundinacea* (L.) Roth, *Pleurospermum uralense* Hoffm., *Rubus saxatilis* L., etc.) is significantly lower in the communities growing in habitats with low suitability (43.9%), than in ones located in the habitats with high suitability (63.4%). Of note is that the situation is reversed for the species typical for deciduous forests of the class *Carpino-Fagetea* (*Aegopodium podagraria* L., *Asarum europaeum* L., *Galium odoratum* (L.) Scop., *Lathyrus vernus* (L.) Bernh., *Lonicera xylosteum* L., etc.): the proportion of this group in the floristic composition of investigated forests decreases from 36% in habitats with low suitability to 29.8% in habitats with high suitability. The proportion of species of the boreal forest class *Vaccinio-Piceetea* is low and varies insignificantly among all investigated pine-broadleaf forests.

### 2.3. Change in Potential Range under Climate Change

Figure 5 shows the models of habitat suitability of pine-broadleaf forests under moderate (RCP4.5) and strong (RCP8.5) climate change scenarios in the middle (the 2040–2060s) and the second half (the 2060–2080s) of the 21st century.

#### 2.3.1. Change in Potential Range in the Southern Urals

The main massifs of pine-broadleaf forests of the suballiance *Tilio-Pinenion* and most of their habitats with high and medium suitability are located in the Southern Ural, which is heterogeneous in terms of landscape and forest vegetation, including deciduous (i), pine-broadleaf (ii) and pine-larch coniferous forests (iii) (Figure 6). The distribution area of pine-broadleaf forests is located in the middle-elevated part of the SU in the contact zone between broadleaf forests of the western slope and the coniferous forests of the eastern slope of the Ural Mountains. Figure 6 shows trends in habitat suitability in the distribution areas of the three above-mentioned forest types in the SU under different climate change scenarios relative to the present time. The orange color shows a decrease and the purple color shows an increase in habitat suitability.

##### Moderate Climate Change Scenario (RCP4.5)

By the middle of the 21st century, in the distribution area of pine-broadleaf forests, the trend of increasing habitat conditions’ suitability will prevail. The total area of all suitable sites will increase slightly (by 5%), and the area of habitats with high suitability will increase significantly (by 41%) (Table 1). The area of habitats with low suitability will be reduced due to their transformation into habitats with medium and high suitability.

In the area of distribution of modern deciduous forests, by the middle of the 21st century, the trend of decreasing habitat suitability for pine-broadleaf forests will prevail, relative to the current situation. The total area of suitable habitats of pine-broadleaf forests will increase slightly, but the mean suitability of habitat conditions will decrease. The proportion of the habitats with high suitability will decrease due to their transformation into the habitats with medium and low suitability.

Habitat suitability will increase throughout the all of the distribution areas of pine-larch coniferous forests by the mid-century. The total area of habitats with high suitability will increase by 3 times, and the total area of habitat with medium suitability will increase by 1.5 times.

By the second half of the 21st century, under the RCP4.5 scenario, the distribution areas of pine-broadleaf and deciduous forests will absolutely dominate sites, where there will be a decrease in habitat suitability. The total area of suitable habitats will decrease slightly relative to the current situation, and habitats with high suitability will almost disappear. In the distribution area of pine-larch coniferous forest, the total area of suitable habitats will increase, but the area of habitats with high suitability will decrease.

##### Strong Climate Change Scenario (RCP8.5)

In the distribution area of modern deciduous forests, the area of habitats with high suitability will decrease by as soon as the middle of the 21st century. On the contrary, the areas of habitats with low and medium suitability will increase due to the conversion of some sites with currently unsuitable habitat conditions to the habitats with medium and low suitability. 

In the distribution area of pine-broadleaf forest, by the middle of the 21st century, the areas of high and medium habitat suitability will increase. The habitats with higher suitability compared to modern ones will prevail. By the second half of the 21st century, under the RCP8.5 scenario, distribution areas of pine-broadleaf and deciduous forests will be dominated by sites where there has been a decline in the habitat suitability. The proportion of habitats with high suitability will decrease, in contrast to medium and low suitability, which will increase due to the conversion from habitats with high suitability.

In the distribution area of pine-larch coniferous forests, the area of suitable habitats will increase by 47% by the middle of the 21st century. In the second half of the 21st century, the area of suitable habitats will slightly decrease, but will still be 41% larger than at present. The area of habitats with high suitability will also decrease slightly compared to the middle of the 21st century, but will still be 2.5 times more than what is currently available.

#### 2.3.2. Change in Potential Range outside the Main Distribution Area of Pine-Broadleaf Forests of the Suballiance *Tilio-Pinenion*

The possible changes in the habitat suitability are expressed to different degrees and directions in the different parts of the pine-broadleaf forests range. In the Middle Urals and the Volga Upland, the areas with suitable habitat conditions are expected to increase by the middle of the 21st century and decrease in the second half of the century under both moderate (RCP4.5) and strong (RCP8.5) scenarios of climate change (Table 1). In the Zhigulevsky Nature Reserve and Samarskaya Luka National Park (the Volga Lowland, Samara region), the area of habitats with high suitability will increase significantly by the middle of the 21st century and then decrease, returning to the category of medium suitability by the second half of the 21st century, under a moderate climate change scenario. Under strong climate change (RCP8.5), the habitat suitability will decrease in the Volga Lowland by the middle of the 21st century. Overall, most habitats with high suitability in all parts of the potential range will be reduced significantly by the second half of the 21st century under both scenarios (RCP4.5 and RCP8.5).

### 2.4. Changes in Climatic Indicators of the Habitats of Pine-Broadleaf Forests under Climate Change

In order to explain the changes in habitat suitability of pine-broadleaf forests under different climate change scenarios, the significant climatic variables were analyzed (Table 2).

Table 2 shows that mean annual temperature in the area of the potential range of pine-broadleaf forests will increase by 2.4° by the middle of the 21st century, and 2.9° in the second half of the 21st century under the moderate climate change scenario (RSP4.5). Under strong climate change (RSP8.5), the increase in mean annual temperature will be, on average, 3.0° by middle of the 21st century and 4.6° in the second half of the 21st century. The minimum temperature of the coldest month will increase in the second half of the 21st century by an average of 3.7° under RSP4.5, and by 5.8 under RSP8.5. The average temperature of the wettest quarter will vary unevenly in the different parts of these forests range.

The average annual precipitation will increase in the second half of the 21st century by an average of 56 mm/year under RSP4.5 and by 60 mm/year under RSP8.5. The precipitation increase will be uneven. In the second half of the 21st century, under climate change scenario RSP4.5, the increase in precipitation will be 63–70 mm/year in the Southern Ural and only 38–49 mm/year in the Bugulma-Belebey Upland, Volga Lowland and Volga Upland. Also, the average precipitation in the driest month will increase throughout all regions. The seasonality of temperature and precipitation will decrease under both climate change scenarios.

## 3. Discussion

### 3.1. Current Potential Range

Temperate mountain forests are very sensitive to climatic changes [52,53,54,55,56], particularly those including tree species growing near the border range. A number of studies in the SU have noted changes in the distribution boundaries of tree species [12,43,44,45], including shifts in the range boundaries of broad-leaved tree species [46]. The increasing role of broad-leaved tree species occurs in the zone of distribution of ecotone communities with the participation of these species. The expansion of ecotone communities occurs in areas with the most favorable (highly suitable) habitat conditions. In some areas currently occupied by ecotone forest communities, in habitats with low suitable conditions, climate change may promote the formation of stands of species which are better adapted to new climatic conditions. The current potential range of pine-broadleaf forests reflects these processes, which occur both in the present and in the past as a result of previous climatic changes.

The potential range of the pine-broadleaf ecotone forests is much wider than their current distribution. According to modeling, most of the pine-broadleaf ecotone forests growing in the habitats with high suitability are located in the SU, which presents the transitional zone between pine-larch coniferous and deciduous forests and is favorable for the preservation of their floristic composition. However, these forests are almost absent on the western slope of the SU, even in the sites with highly suitable habitat conditions, due to forestry activities (selective cutting of pine trees) and their replacement by deciduous forests [57].

On the Bugulma-Belebey Upland, pine-broadleaved forests are not widespread due to the lack of habitats with high and medium suitability. Rare small areas of pine-broadleaf forests usually occur in the sites unsuitable for agricultural use, i.e., slopes, ravines, etc. 

The Volga Upland is the southern range border of *Pinus sylvestris*, which has formed here boreal and hemiboreal pine forests [58,59]. In the Volga Upland, pine-broadleaf forests occupy small areas in habitats atypical for Southern Ural pine-broadleaf forests. These forests grow on sandy and sandy loam sediments of the Paleogene, ancient alluvial sands and chalk substrates. These forests are usually located in the lower parts of slopes, in ravines and on steep river banks in the habitats with low, rarely medium suitability. In the central part of the Volga Upland in the Ulyanovsk region, there are pine-broadleaf forests with a pine upper layer and the undergrowth is formed by broad-leaved trees. These forests were widely distributed in the past, and the reduction in their current range is associated with forestry activities, since the felling initiates the transformation of this forest type into deciduous [59]. In the northwestern part of the Volga Upland in the Republic of Mordovia and Penza region, these forests are located in river valleys and differ from the pine-broadleaf forests of the SU in their greater representation of boreal species in the undergrowth [60,61]. In the south of the Volga Upland in the Saratov region, these forests are less widespread and occur in the Volga River floodplain [58]. 

In contrast to the main territory of the Volga Upland, the habitats with high suitability are found on its border with the Volga Lowland in the Samara region (in the Zhigulevsky Nature Reserve and the Samarskaya Luka National Park). In this area, pine-broadleaf forests are located on flat sites and form the ecotone vegetation between mountain mossy or herb-rich pine forests and thermophilous oak forests. The floristic composition of these forests is characterized by a high proportion of steppe species [62].

All pine-broadleaf forests of the Volga Upland and Volga Lowland were ordered to the suballiance *Querco robori-Tilienion cordatae* Morozova 2016 (class *Carpino-Fagetea sylvaticae* Jakucs ex Passarge 1968, order *Carpinetalia betuli* P. Fukarek 1968, alliance *Querco roboris-Tilion cordatae* Solomeshch et Laiviņš ex Bulokhov et Solomeshch in Bulokhov et Semenishchenkov) [63] and represent a vicarious type, in relation to the pine-broadleaf forests of the SU. The main floristic differences between two suballiances of pine-broadleaf forests are the presence of *Corylus avellana* L. and the greater participation of *Quercus robur,* boreal and steppe species in the communities of the suballiance *Querco-Tilienion*, as well as greater participation of hemiboreal species in the communities of the suballiance *Tilio-Pinenion*. These differences may be explained by both warmer and arid climatic conditions and intensive repeated cutting of pine trees, which caused an impoverishment of floristic diversity. Nevertheless, the hemiboreal forest species *Chamaecytisus ruthenicus* (Fisch. ex Woloszcz.) Klásk., *Carex rhizina* Blytt ex Lindblom, *Viola collina* Besser, *Brachypodium pinnatum* (L.) Beauv., *Solidago virgaurea* L. and *Hieracium umbellatum* L., indicating the common historical past of these two suballiances, are sporadically presented in the communities of the suballiance *Querco-Tilienion*. 

Differentiation of forest vegetation of these suballiances occurred by two limiting factors, i.e., the climate aridization in the Volga Upland and extremely low winter temperatures in the SU. The dieback of oak trees in the SU during the 20th century, as a result of extreme climatic events and outbreaks of the Gypsy moth (*Lymantria dispar* L.), increased the differences between vegetation of these suballiances [64,65]. Also, in the SU, several outbreaks of “Dutch elm disease” during the second half of the 20th century caused mass dieback of *Ulmus glabra* [66,67]. Also, spring frosts during budbreak, or the formation of young immature leaves and branches, had a significant impact on the health of broad-leaved trees, not only on oak and elm, but also on maple (*Acer platanoides*), which is the early-vegetating tree species [37,38,39,40]. Under conditions of climate warming in the SU, abnormal winter frosts have ceased and the frequency of spring frosts has decreased at the current time [68]. This significantly improved the growing conditions of broad-leaved trees in mixed pine-broadleaf forests near the eastern border of their range. At present, an increase in the seed production of linden, maple and elm has been revealed. The main transfer of these species’ seeds occurs from the warmed upper parts of the mountain slopes to the lower parts, where deciduous trees form small undergrowth and create the opportunity to penetrate into large undergrowth and the upper tree layer during climate warming. *Acer platanoides* has the most efficient seed reproduction in the SU. As in the case of *Acer rubrum* L. in European forests, this species penetrates into the tree layer due to the appearance of single trees in the tree stand gaps, as well as from the undergrowth that survived the period of suppression and was preserved during felling [69]. The selective cutting of conifers and bacterial wetwood of silver birch (*Erwinia multivora* Sch.-Parf.) that may advance to the middle part of the SU under climate warming can also be favorable for the spreading of broad-leaved trees.

The analysis of differences in floristic composition of pine-broadleaf forests of the suballiance *Tilio-Pinenion* depending on the degree of habitat suitability has shown that pine-broadleaf forests located in habitats with low suitability in the SU are characterized by a lower proportion of hemiboreal species and a higher proportion of species characteristic of deciduous forests. That is, in their floristic composition, they are closer to the pine-broadleaf forests of the Volga Upland and Volga Lowland. However, these forests also differ from pine-broadleaf forests of the Volga Upland and Volga Lowland by a smaller proportion of *Quercus robur* in the stand and the absence of *Corylus avellana* L., whose range does not reach the Ural Mountains. The selective cutting of pine in pine-broadleaf forests leads to the formation of linden forests or maple forests in the SU [70] and to secondary oak forests in the Volga Upland [58].

Thus, overruns of the potential range of pine-broadleaf forests over their real distribution can be explained by two main reasons: (1) in warm and arid climate conditions of the Volga Upland and Volga Lowland, the forests of the suballiance *Tilio-Pinenion* have been replaced by forests of the suballiance *Querco robori-Tilienion cordatae*; (2) in the western slope of the SU, the areas of forests of the suballiance *Tilio-Pinenion* were drastically reduced due to forestry activities.

### 3.2. Change in Potential Range of Pine-Broadleaf Forests under Climate Change

Projected climate change under both the RCP4.5 and RCP8.5 scenarios will result in significant changes in growing conditions of pine-broadleaf forest habitats both in the SU, which is the main distribution area of these forests, and outside.

#### 3.2.1. Change in Potential Range of Pine-Broadleaf Forests in the Southern Ural Region

On the Ufa Plateau, habitat suitability for pine-broadleaf forests is projected to decrease under both RCP4.5 and RCP8.5 scenarios. The decrease in habitat suitability is associated with a projected increase in temperature and precipitation in winter months and a decrease in climate continentality. Therefore, in this area, a gradual replacement of pine-broadleaf forests by deciduous forests is predicted. These changes are accelerated by selective felling of pine trees.

The current average annual temperature in the SU decreases along the distribution gradient of deciduous forests, through pine-broadleaf forests and towards pine-larch coniferous forests, and is 3.2, 2.3 and 2.0 °C, respectively. In the modern area of distribution of pine-larch coniferous forests, climate warming will cause the temperature shift close to the optimum for pine-broadleaf forests in the middle of the 21st century, and then, in the second half of the 21st century, the habitat suitability will worsen due to aridification. 

In the current area of deciduous forests, habitat suitability will decrease due to global warming and a decrease in climate continentality, causing a gradual replacement of mixed forests with deciduous forests. This trend may be accelerated by selective felling of pine [56,63]. For the same reasons, in the current distribution area of pine-broadleaf forest, the habitats’ suitability will increase significantly by the middle of the 21st century and will gradually decrease in the second half of the 21st century. Thus, in the SU, climatic change will cause the penetration of broad-leaved species into coniferous forests and the shift of the potential range of pine-broadleaf forests to the east. In the second half of the 21st century, the increasing aridization of the climate will shift the potential range border of pine-broadleaf forests back to the west due to their gradual replacement by hemiboreal pine-larch forests.

#### 3.2.2. Change in Potential Range of Pine-Broadleaf Forests outside the Main Distribution Area of the Forests of the Suballiance *Tilio-Pinenion*

In the Volga Upland and Volga Lowland, under both moderate (RCP4.5) and strong (RCP8.5) climate change scenarios, a strong increase in mean annual temperature is expected together with relatively small changes in mean annual and winter precipitation, which will cause an increase in transpiration and evaporation of soil moisture. This, in turn, can intensify the processes of climate aridification and the replacement of pine-broadleaf forests by xerophytic pine forests or unproductive oak forests, depending on the relief features.

In the Middle Urals, the current average annual temperature is only 0.5 °C, and the minimum average monthly temperature of the coldest month is the lowest in the potential area of pine-broadleaf forests. Global warming will shift the temperature regime close to optimal conditions for pine-broadleaf forests; however, the expected increase in the average annual precipitation will favor the distribution of dark coniferous and mixed forests with *Picea obovata* and *Abies sibirica* (class *Asaro europaei–Abietetea sibiricae* Ermakov, Mucina et Zhitlukhina in Willner et al. 2016, order *Abietetalia sibiricae* (Ermakov in Ermakov et al. 2000) Ermakov 2006, alliance *Aconito septentrionalis-Piceion obovatae Solomeshch*, Grigoriev, Khaziakhmetov et Baisheva in Martynenko et al. 2008) [4].

Modeling results predict only changes in habitat conditions for pine-broadleaf forests. It should be taken into account that changes in vegetation cover in these forests are slower than changes in habitat suitability [47]. This is due to the longevity of tree species and limitations in seed dispersal. Nevertheless, in the SU, the eastward advance of broad-leaved species over the last 40 years has averaged 10–15 km [46]. With the acceleration of climatic changes, especially in combination with anthropogenic impact, these processes may dramatically accelerate.

In order to protect the rare types of pine-broadleaf forests in the SU, it is necessary to lift the ban on the cutting of broad-leaved tree species at the border of their ranges, as selective cutting of pine trees leads to replacement of these forests with deciduous ones. With an increase in the area of low productive deciduous stands replacing pine-broad-leaved forests, there will be a need to create artificial pine plantations.

## 4. Materials and Methods

The maximum entropy modeling software (MaxEnt v3.4.4) was used to assess the changes in habitat suitability of pine-broadleaf forests of the suballiance *Tilio-Pinenion* under the climatic changes [71,72,73]. The data on 120 geo-referenced points (localities of geobotanical relevés performed by the authors of the manuscript during 2010–2023 and classified according to the Brown-Blanquet approach) were used for modeling (Figure 7). Points with a minimum distance of 1 km between them were used in the modeling.

Two scenarios of further climate change were selected for modeling: moderate climate change RCP4.5 and strong climate change RCP8.5 scenarios developed under the IPCC (Intergovernmental Panel on Climate Change) project [74]. Moderate climate change is a scenario of stabilizing climate change through policies to reduce greenhouse gas emissions [74,75,76]. Under this scenario, temperatures are projected to increase by 1.4 °C in the middle of the 21st century (2040–2060s) and by 1.8 °C in the second half of the 21st century (2061–2080s) [77]. Under the RCP8.5 scenario, temperatures are projected to increase by 2 °C in the middle of the 21st century and 3.7 °C in the second half of the 21st century [78]. At present, the analysis of climate change dynamics indicates the realization of a moderate climate change scenario RCP4.5, but the possibility of more severe climate change cannot be completely ruled out [78]. To predict the future distribution of the pine-broadleaf forest, we used an ensemble of four climate change models as recommended by McSweeney et al. [79]: CCSM4 [80], NorESM1-M [81], MIROC-ESM [82], INMCM4 [83], for which the corresponding BIOCLIM datasets were used (CHELSA). We used the following MaxEnt settings: maximum iterations—1000, replicates—5 and output format—“cloglog”. The “cloglog” output gives a score between 0 (completely unsuitable) and 1 (completely consistent with the species or community optimum) of habitat suitability.

As predictors in the modeling, we used the set of Bioclim climatic variables from the CHELSA database (Climatologies at high resolution for the earth’s land surface areas) [84,85], and the digital elevation model GMTED2010 [86] with a 30 arc sec resolution. The elevation difference was used as a rough estimate of slope steepness. Soil raster maps of the global digital soil mapping system SoilGrids were used to characterize soils of pine-broadleaf forest habitats [87,88]. Raster layers of environmental data were restricted to Eurasia. We generated a Pearson correlation matrix of environmental predictors, and in the case of a correlation coefficient greater than or equal to 0.8, one of the variables was excluded to prevent multicollinearity and model overfitting [89] (Appendix A). 

Table 3 shows the contribution of the climatic and soil variables to the model. Four environmental variables had the highest contribution: precipitation amount of the driest month (Bio14), temperature seasonality (standard deviation of the monthly mean temperatures) (Bio4), the elevation difference within one pixel (h_max−min_), and mean daily minimum air temperature of the coldest month (Bio6).

The AUC indicator was used for the statistical evaluation of the model [50]. Standard MaxEnt tests (jackknife, permutation importance and percent contribution) were used to assess the reliability of the predictors’ contribution to the model. We applied the “Maximum test sensitivity plus specificity” threshold as the lowest limit for habitat suitability [90]. In the final models, the habitat suitability with values above the lowest limit for habitat suitability was divided into three equal groups: low, medium, and high. The area covered by each suitability level was calculated using QGIS v3.14.

Based on the 178 complete relevés, a phytosociological spectrum of communities was compiled, intended to analyze changes in floristic composition of pine-broadleaf forests depending on habitat suitability. For this purpose, a buffer of 10 × 10 m was created for each geo-referenced point (relevé locality). The polygons were superimposed on the model of potential range and habitat suitability in these localities was calculated using the “Zonal Statistics” module in QGIS. All herbaceous species mentioned in geobotanical relevés have been assigned to vegetation classes in whose communities they have an ecological optimum. After this, the proportion of each species in the projective cover of the herbaceous layer was calculated taking into account the constancy (percentage of occurrence in the group of relevés) and abundance of species (projective cover on a sample plot). The proportion of broad-leaved tree species in the understorey was also calculated similarly.

## 5. Conclusions

An analysis of modeling has shown that the interpretation of potential ranges of plant communities should be carried out with caution as the habitats of vicariant plant communities are difficult to separate. In spite of the differences in the floristic composition of pine-broadleaf forests of the SU and the Volga Upland, the presence of common species indicates a close historical past of the forests of these two suballiances. 

Possible changes in the range borders of broad-leaved trees can be estimated by the distance between the isolated habitats of these species from their main range [91,92,93]. The modern potential range of broad-leaved trees includes these isolated habitats. This indicates the potential spread of broad-leaved species to the maximum eastern limits in the Holocene. However, the range borders of trees change more slowly than climate [47]. In addition, these processes will depend on the particular climate changes. Monitoring using permanent sample plots and methods of remote sensing are needed to control the rate of change of floristic composition and distribution of ecotone communities.

Understanding the role of anthropogenic and climatic factors in vegetation change should help regional environmental management and is necessary to address reforestation and conservation of forest ecosystems.

## Figures and Tables

**Figure 1 plants-12-03698-f001:**
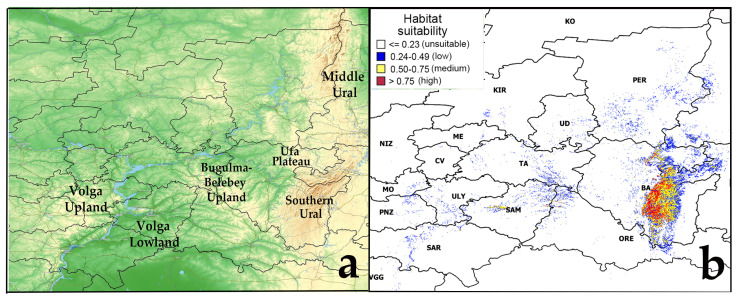
Current potential range of pine-broadleaf forests of suballiance Tilio-Pinenion. Abbreviated names of Russian regions are given according to the International Organization for Standardization ISO 3166-1 [51]: BA—Republic of Bashkortostan, CV—Chuvash Republic, KIR—Kirov region, ME—Republic of Mari El, MO—Republic of Mordovia, NIZ—Nizhny Novgorod region, ORE—Orenburg region, PER—Perm Krai, PNZ—Penza region, SAM—Samara region, SAR—Saratov region, TA—Republic of Tatarstan, UD—Udmurt Republic, ULY—Ulyanovsk region, VGG—Volgograd region. Note: (**a**)—physical map of the potential distribution area, (**b**)—potential range of pine-broadleaf forests.

**Figure 2 plants-12-03698-f002:**
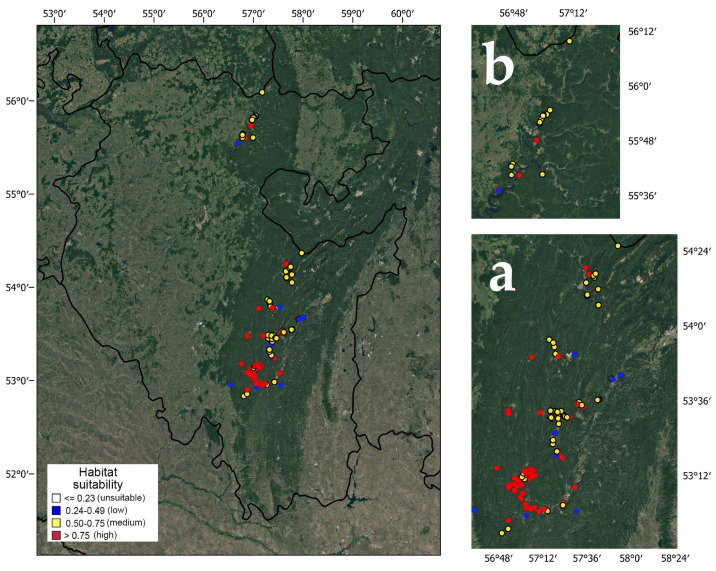
Habitat suitability in pine-broadleaf forests of suballiance *Tilio-Pinenion* in the Southern Ural (**a**) and the Ufa Plateau (**b**).

**Figure 3 plants-12-03698-f003:**
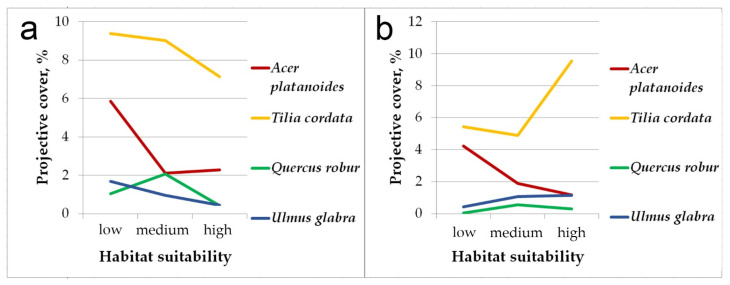
The differences in total projective cover of broad-leaved species in the pine-broadleaf forests located in the habitats with different suitability: (**a**)—general tree layer and large undergrowth, (**b**)—small undergrowth.

**Figure 4 plants-12-03698-f004:**
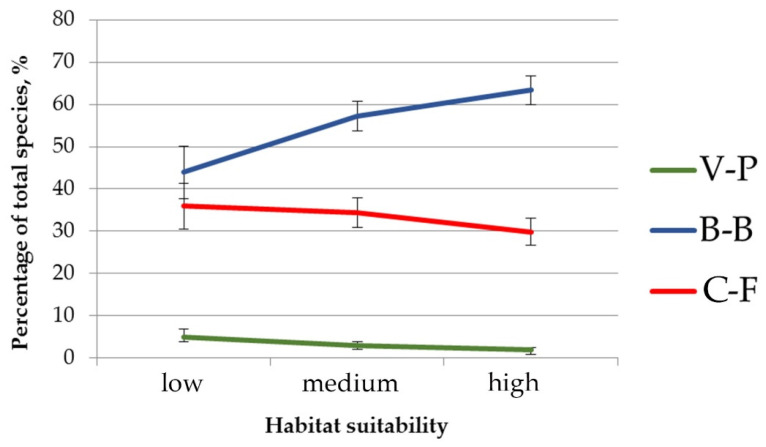
The proportion of species typical for different vegetation classes in the herb layer of pine-broadleaf forests growing in the habitats with different degrees of suitability. Note: C-F—mesic deciduous and mixed forests of the class *Carpino-Fagetea*, B-B—hemiboreal pine and birch-pine herb-rich open forests of the class *Brachypodio-Betuletea*, V-P—coniferous boreal taiga forests of the class *Vaccinio-Piceetea*.

**Figure 5 plants-12-03698-f005:**
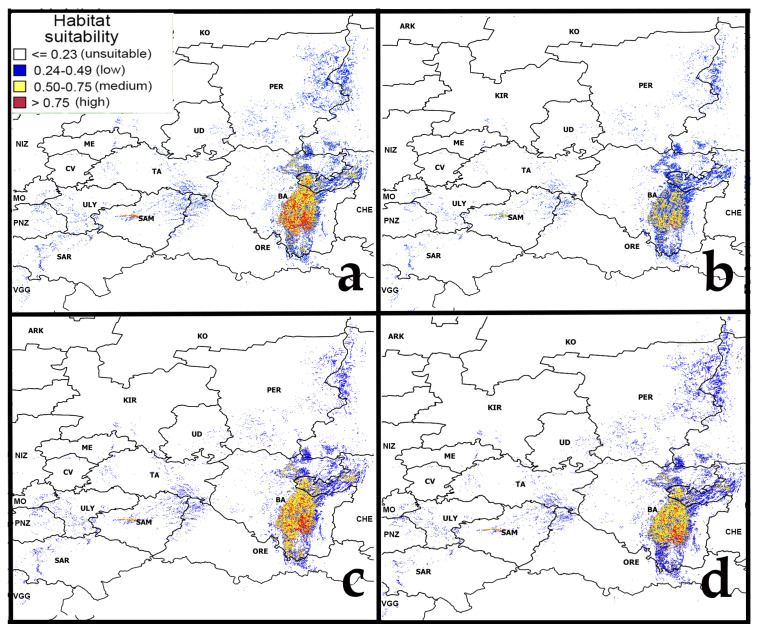
Forecast of habitat suitability of pine-broadleaf forests under moderate and strong climate change. Note: (**a**)—RCP4.5 (2050s 21st Century); (**b**)—RCP4.5 (2070s 21st Century); (**c**)—RCP8.5 (2050s 21st Century); (**d**)—RCP8.5 (2070s 21st Century). Abbreviated names of regions are shown in the notes of Figure 1.

**Figure 6 plants-12-03698-f006:**
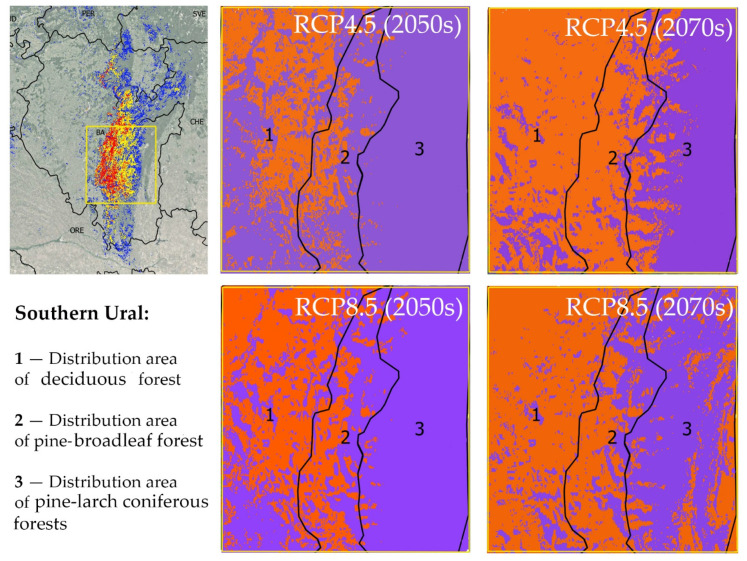
Change in habitat suitability of pine-broadleaf forests of suballiance *Tilio-Pinenion* in the Southern Ural under different climate change scenarios (orange color shows a decrease in habitat suitability and purple color shows an increase in habitat suitability). In the left part, the yellow box indicates the study area in the Southern Ural. On the right side, modeling results for the same area are shown on a larger scale.

**Figure 7 plants-12-03698-f007:**
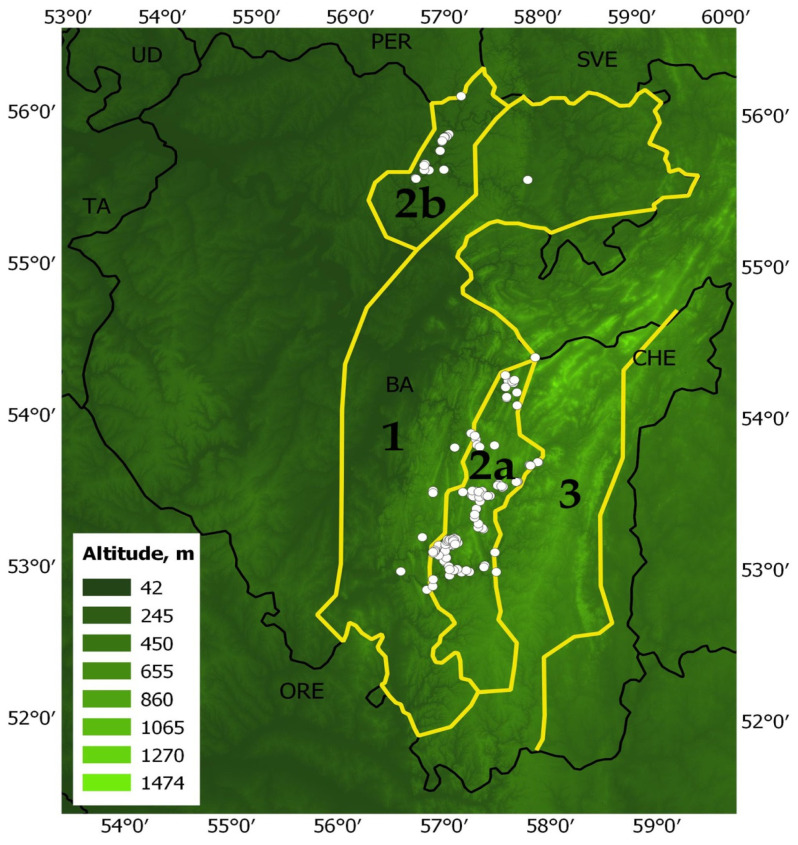
Territory of maximum distribution of pine-broadleaf forests: 1—distribution area of deciduous forests in the Southern Ural; 2a—distribution area of pine-broadleaf forests in the Southern Ural; 2b—distribution area of pine-broadleaf forests on the Ufa Plateau; 3—distribution area of pine-larch coniferous forest in the Southern Ural; white circles are images of forest vegetation.

**Table 1 plants-12-03698-t001:** The regional changes in the areas with different habitat suitability of pine-broadleaf forests under different scenarios of climate change in 2040–2060s and 2060–2080s, as a percentage of the modern areas.

Regions	Changes in Areas under Climate Change, %							
	All Suitability *		Low Suitability		Medium Suitability		High Suitability	
	2040–2060	2060–2080	2040–2060	2060–2080	2040–2060	2060–2080	2040–2060	2060–2080
RCP4.5								
Ufa plateau	−4	−29	2	8	5	−79	−84	−100
Southern Ural (distribution area of pine-broadleaf forests)	5	−3	−16	57	1	1	41	−86
Southern Ural (distribution area of deciduous forest)	13	−4	16	56	40	−8	−30	−97
Southern Ural (distribution area of pine-larch coniferous forests)	36	15	18	31	55	−16	298	−61
Middle Ural	186	46	178	50	421	−63	-	-
Bugulma-Belebey Upland	−4	−51	0	−43	−24	−89	−37	−99
Volga Lowland	−1	−26	−5	−19	−26	−25	199	−96
Volga Upland	19	−37	16	−36	67	−57	483	−100
**RCP8.5**								
Ufa plateau	−6	−8	−6	10	0	−20	−100	−100
Southern Ural (distribution area of pine-broadleaf forests)	5	2	5	6	16	18	12	−39
Southern Ural (distribution area of deciduous forests)	15	7	15	31	75	42	−64	−83
Southern Ural (distribution area of pine-larch coniferous forests)	47	41	47	31	85	53	262	148
Middle Ural	95	179	94	183	103	69	-	-
Bugulma-Belebey Upland	−19	−42	−11	−32	−56	−84	−89	−97
Volga Lowland	−8	−35	−9	−37	−11	−35	19	−23
Volga Upland	21	−23	18	−23	72	−30	167	−100

* sum of different habitat suitabilities.

**Table 2 plants-12-03698-t002:** Environmental parameters of pine-broadleaf forests of suballiance *Tilio-Pinenion* under different climate change scenarios in 2040–2060s and 2060–2080s.

	Ufa Plateau	Southern Ural (Distribution Area of Deciduous Forest)	Southern Ural (Distribution Area of Pine-Broadleaf Forest)	Southern Ural (Distribution Area of Coniferous Forest)	Middle Ural	Bugulma-Belebey Upland	Volga Lowland	Volga Upland
Average annual temperature, °C (bio1)								
Current	3.0	3.2	2.3	2.0	0.5	4.4	**5.3**	**5.3**
RCP4.5 (2040–2060)	5.5	5.7	4.7	4.5	2.9	6.8	**7.6**	**7.5**
RCP4.5 (2060–2080)	6.0	6.1	5.2	4.9	3.4	7.3	**8.1**	**8.0**
RCP8.5 (2040–2060)	6.1	6.3	5.3	5.0	3.7	7.4	**8.3**	**8.1**
RCP8.5 (2060–2080)	7.6	7.8	6.8	6.6	5.3	8.8	**9.6**	**9.5**
Minimum temperature of the coldest month, °C (bio6)								
Current	−16.0	−16.2	−17.4	−17.7	−19.0	−15.0	**−14.2**	**−13.4**
RCP4.5 (2040–2060)	−13.0	−13.2	−14.4	−14.7	−15.8	−11.9	**−11.0**	**−10.3**
RCP4.5 (2060–2080)	−12.4	−12.5	−13.5	−13.9	−15.5	−11.4	**−10.5**	**−9.9**
RCP8.5 (2040–2060)	−12.6	−13.1	−14.4	−14.6	−14.9	−11.7	**−10.8**	**−10.2**
RCP8.5 (2060–2080)	−10.1	−10.4	−11.6	−11.9	−12.7	−9.5	**−8.8**	**−8.3**
Average temperature of the wettest quarter, °C (bio8)								
Current	16.0	16.9	16.9	17.0	15.3	19.8	20.5	16.6
RCP4.5 (2040–2060)	14.1	16.3	16.8	18.1	16.6	20.0	20.8	18.4
RCP4.5 (2060–2080)	13.7	15.9	15.7	18.2	15.7	19.9	19.5	17.1
RCP8.5 (2040–2060)	13.8	16.7	16.9	19.1	14.7	18.1	19.2	16.9
RCP8.5 (2060–2080)	17.0	16.6	16.2	19.6	18.7	19.1	17.4	16.4
Temperature seasonality (bio4)								
Current	11,454	11,698	11,826	11,735	11,443	11,694	11,580	11,143
RCP4.5 (2040–2060)	11,259	11,476	11,568	11,492	11,341	11,456	11,379	10,993
RCP4.5 (2060–2080)	11,003	11,213	11,300	11,218	11,145	11,220	11,154	10,781
RCP8.5 (2040–2060)	11,265	11,539	11,684	11,576	11,147	11,530	11,448	11,076
RCP8.5 (2060–2080)	10,827	11,091	11,220	11,124	10,869	11,076	11,031	10,715
Average annual precipitation, mm/year (bio12)								
Current	613.9	596.5	593.8	531.0	**650.0**	474.0	479.3	525.3
RCP4.5 (2040–2060)	665.4	646.3	644.1	575.4	**700.5**	507.0	512.9	561.6
RCP4.5 (2060–2080)	676.7	660.9	663.6	588.5	**698.0**	523.5	524.6	562.9
RCP8.5 (2040–2060)	631.3	609.4	605.4	541.0	**675.4**	486.1	491.3	537.1
RCP8.5 (2060–2080)	692.5	667.0	659.9	590.3	**733.1**	525.3	527.5	568.2
Precipitation of driest month, mm (bio14)								
Current	27.4	26.7	28.1	22.6	26.7	23.1	25.9	27.8
RCP4.5 (2040–2060)	30.1	28.8	30.2	24.6	30.0	23.7	26.0	27.7
RCP4.5 (2060–2080)	31.6	30.0	30.6	25.0	30.1	24.5	27.2	28.8
RCP8.5 (2040–2060)	30.9	29.0	29.2	23.6	28.7	23.6	25.9	27.1
RCP8.5 (2060–2080)	31.6	31.4	33.6	27.1	31.2	26.6	29.5	30.2
Seasonality of precipitation (bio15)								
Current	32.1	32.5	31.3	38.7	37.3	28.9	22.2	25.5
RCP4.5 (2040–2060)	30.2	31.2	30.6	37.0	34.3	27.8	22.8	27.4
RCP4.5 (2060–2080)	30.1	31.2	30.8	36.3	34.1	29.8	24.0	25.2
RCP8.5 (2040–2060)	28.6	29.2	28.6	35.2	32.9	27.7	22.1	24.6
RCP8.5 (2060–2080)	28.1	28.5	27.0	33.4	32.7	25.0	19.7	24.0

Maximum values of climatic indicators are marked in bold, minimum values of climatic indicators are underlined.

**Table 3 plants-12-03698-t003:** Contribution of environmental variables to the model of the potential range of pine-broadleaf forests of the suballiance *Tilio-Pinenion*.

Code	Environmental Variable	Percent Contribution	Permutation Importance
Bio14	Precipitation amount of the driest month	31.3	3.2
Bio4	Temperature seasonality	28.7	64.4
h_max−min_	Difference between maximum and minimum elevation, m	24.5	5.7
Bio6	Mean daily minimum air temperature of the coldest month	10.9	13.4
nitrogen 0–5 sm	Total nitrogen in the upper 0–5 cm soil layer	1.8	7.6
Bio8	Mean daily mean air temperatures of the wettest quarter	1.1	0.4
soc 0–5 sm	Soil organic carbon content in the fine earth fraction in the upper 0–5 cm soil layer	0.8	0.6
soil	Soil stoniness (Volumetric fraction of coarse fragments (> 2 mm) [sm^3^/dm^3^])	0.5	1.8
Bio15	Precipitation seasonality	0.2	2.7

## Data Availability

The data that support the findings of this study are available from the corresponding authors upon reasonable request.

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
