# Peer review of "Analysis of the Potential Range of Mountain Pine-Broadleaf Ecotone Forests and Its Changes under Moderate and Strong Climate Change in the 21st Century"

_plants, 2023, doi:10.3390/plants12213698_

Round 1

Reviewer 1 Report

Review of Plants-2613237

This manuscript describes a MaxEnt analysis of the potential range shift of pine-broadleaf forests in the Ural Mountains. These types of analyses are very useful for both the broader scientific community and land managers to understand how climate change will impact landscapes of interest. The analysis is fairly straightforward and easy to comprehend, but the structure of the paper introduces some confusion that makes it hard for the reader to draw clear conclusions.

PRIMARY COMMENTS:

Three major suggestions that can improve the manuscript are 1) consistent terminology, 2) simplified discussion, and 3) parallel structure.

1) For example, sometimes the authors refer to “pine-broadleaf”, etc. and sometimes as V-P, B-B, C-F (eg Figures 3 and 4). Also, text on Lines 199-200 say “deciduous (i), pine-broadleaf (ii) and pine-larch coniferous forests (iii)” in referring to Figure 6, which says “broad-leaved forest”, “pine-broad-leaved forest” and “coniferous forest”. These are two of many examples.

2) The authors jump around between regions a lot in ways that are hard to track. Several figures and most of the Results are focused on the Southern Ural region (with the Ufa plateau also featured frequently), while many tables and the Discussion covers a lot of regions. It is very difficult to keep track of the significance of the different areas.

3) One thing that can improve the readability (especially point 2 above) would be a more parallel structure. For example, ordering the Results sections in a consistent order, and use the same order in the Discussion. Right now, in both cases they jump around a lot from paragraph to paragraph, sometimes in order by cover type (pine-broadleaf, pine, broadleaf), sometimes by region (seemingly in random order), sometimes by analysis type.

ADDITIONAL COMMENTS:

Clarifying the difference between “current potential range” and “change of potential range under climate change” would be helpful. I assume the “current potential range” is suitability for pine-broadleaf forests under current climatic conditions? It’s hard to interpret that without seeing where pine-broadleaf forests occur now… which I think is what Figure 2 is providing at a point-scale, whereas Figure 1 just shows the potential range irrespective of the where it occurs now. And then Figure 5 is the range under climate scenarios, but Figures 1 and 5 look so similar that it would help the reader to make the differences very obvious. I guess overall, what each of these figures (and other figures) is providing is hard to keep straight. Suggestion 3 above will likely help a lot.

Table 2 is hard to interpret, and may be better suited as a multi-panel set of bar graphs. Also the text in the results only discusses the Southern Ural data, so the reader is left wondering what the significance of the other data is. Consider explaining the other region’s data in text, or simplifying this table/figure and putting the rest in a supplement.

Table 3 is also too much data. Consider a small table, or bolding/highlighting important data.

Figure 3 is mis-labelled as Figure 2.

Figures 3 and 4: label axes as low, medium, and high and avoid the 1,2, 3 altogether.

Lines 399-402: How are these models related to the RCP models? It is confusing as the results show RCP models and don’t reference these terms – we never see these terms again.

Lines 404-408: What is the spatial coverage of these variables – global? Regional? Knowing more about how they were derived would help interpretation. Also, because you don’t list the actual variables here, reference Table 1 – the reader wants to know the specific variables.

Lines 416-417: How are low, medium, and high defined?

Author Response

This manuscript describes a MaxEnt analysis of the potential range shift of pine-broadleaf forests in the Ural Mountains. These types of analyses are very useful for both the broader scientific community and land managers to understand how climate change will impact landscapes of interest. The analysis is fairly straightforward and easy to comprehend, but the structure of the paper introduces some confusion that makes it hard for the reader to draw clear conclusions.

Answer: Dear Reviewer. We are thank you for careful review of our article. Your recommendations and comments are very useful and improved the quality of manuscript. Below are the answers to your comments (in the text of article corrections are marked in red color).

PRIMARY COMMENTS:

Three major suggestions that can improve the manuscript are 1) consistent terminology, 2) simplified discussion, and 3) parallel structure.

1) For example, sometimes the authors refer to “pine-broadleaf”, etc. and sometimes as V-P, B-B, C-F (eg Figures 3 and 4). Also, text on Lines 199-200 say “deciduous (i), pine-broadleaf (ii) and pine-larch coniferous forests (iii)” in referring to Figure 6, which says “broad-leaved forest”, “pine-broad-leaved forest” and “coniferous forest”. These are two of many examples.

Answer: Acknowledged. All terms are consistent.

2) The authors jump around between regions a lot in ways that are hard to track. Several figures and most of the Results are focused on the Southern Ural region (with the Ufa plateau also featured frequently), while many tables and the Discussion covers a lot of regions. It is very difficult to keep track of the significance of the different areas.

Answer: We have added a map with the areas mentioned to make the text easier to understand.

3) One thing that can improve the readability (especially point 2 above) would be a more parallel structure. For example, ordering the Results sections in a consistent order, and use the same order in the Discussion. Right now, in both cases they jump around a lot from paragraph to paragraph, sometimes in order by cover type (pine-broadleaf, pine, broadleaf), sometimes by region (seemingly in random order), sometimes by analysis type.

Answer: Thanks for comment. We have tried to improve the structure of our article and make some explanations in the text.

Clarifying the difference between “current potential range” and “change of potential range under climate change” would be helpful. I assume the “current potential range” is suitability for pine-broadleaf forests under current climatic conditions? It’s hard to interpret that without seeing where pine-broadleaf forests occur now… which I think is what Figure 2 is providing at a point-scale, whereas Figure 1 just shows the potential range irrespective of the where it occurs now. And then Figure 5 is the range under climate scenarios, but Figures 1 and 5 look so similar that it would help the reader to make the differences very obvious. I guess overall, what each of these figures (and other figures) is providing is hard to keep straight. Suggestion 3 above will likely help a lot.

Answer: The current potential range reflects the distribution of habitats with suitable conditions for these forests under current climatic conditions. We have added the necessary explanations to the text.

Table 2 is hard to interpret, and may be better suited as a multi-panel set of bar graphs. Also the text in the results only discusses the Southern Ural data, so the reader is left wondering what the significance of the other data is. Consider explaining the other region’s data in text, or simplifying this table/figure and putting the rest in a supplement.

Thanks for the suggestion. In our case, the histogram will be uninformative because the changes range from 1 to 421%. As a result, relatively small changes will visually merge with the axis.

Table 3 is also too much data. Consider a small table, or bolding/highlighting important data.

Answer: We completely agree with your comment. We marked the most important data discussed in the text in bold to make the table easier to navigate.

Figure 3 is mis-labelled as Figure 2.

Thank you! Done.

Figures 3 and 4: label axes as low, medium, and high and avoid the 1,2, 3 altogether.

Thank you! Done.

Lines 399-402: How are these models related to the RCP models? It is confusing as the results show RCP models and don’t reference these terms – we never see these terms again.

Thanks for the comment! Corrections have been made both here and further down the text.

Lines 404-408: What is the spatial coverage of these variables – global? Regional? Knowing more about how they were derived would help interpretation. Also, because you don’t list the actual variables here, reference Table 1 – the reader wants to know the specific variables.

The raster layers of the ecological data were limited to Eurasia. The illustrations show only part of the area where potential pine-broadleaf forest habitats were identified. The text is supplemented with the necessary explanations.

Lines 416-417: How are low, medium, and high defined?

We applied the “Maximum test sensitivity plus specificity” threshold as the lowest limit for habitat suitability. In the final models, the habitat suitability with values above the lowest limit for habitat suitability was divided into three equal groups: low, medium, and high. The area covered by each suitability level was calculated using QGIS v3.14. The text is supplemented with the necessary explanations.

Reviewer 2 Report

Comments on plant-2613237-v1

The current study focuses on the impact of climate change on pine-broadleaf ecotone in the Southern Ural region. It aims to define the current potential range of the Tilio-Pinenion sub-alliance and also to model the habitat suitability of studied forests under two different climate change scenarios. The research uses MaxEnt software and data on 120 georeferenced localities and some other data(climate, soil, elevation). 

The Methods part of this study description needs to be considerably supplemented. It is not entirely clear how some results were obtained here due to the deficiencies in this section.

The Discussion should be revised and needs to be considerably improved. Discussion is a crucial part of the research article, where authors should interpret the study's results, relate them to existing literature, and provide insights into the implications of their findings. I would expect to see the results of this study also compared/aligned with other studies to get to know how similar/different the outcome of this study is from other studies. Also, discuss how trustworthy your study results are and explain the implications of your findings. Why should the reader care about your study (theoretical/practical implications)? How might it be used in decision-making, practice, or policy? Provide suggestions for further research based on your findings and discuss limitations in the discussion part but not in the Conclusions. Avoid repeating the result section; interpret the findings and support your claims with evidence from the study and other literature. 

Consider improving the Conclusions part, where you should conclude the findings of your study.

Comments:

L.33, 43 use " in the transitional zone between..." instead of "in the zone of contact."

L. 40 " The role of pine (Pinus sylvestris L.) in the vegetation cover of the SU changed..."- Do you mean "share of pine"? Provide a reference for this statement.

L.44 "Sustainable ecotone mixed pine-broadleaved forests have appeared." what do you mean by "sustainable" here? In what way is it sustainable? Moreover, "have appeared"? Pay attention to grammar: shouldn't it be "ecotone of mixed pine-broadleaved forests"?

L.47-51 Sentence is difficult to follow; consider improving. e.g. "In terms of floristic classification, these ecotone forests fall within the Carpino Fagetea sylvaticae class as designated by Jakucs ex Passarge in 1968. They are categorized under the order Carpinetalia betuli P.Fukarek 1968, within the alliance Aconito lycoctoni Tilion cordatae, which was described by Solomeshch et Grigoriev in Willner et al. in 2016. Furthermore, they are subcategorized under the suballiance Tilio cordatae Pinenion sylvestris, as proposed by Shirokikh et al. in 2021."

L.51-55 unable to understand the sentence. 

Shouldn't it be: "Simulations of the distribution of vegetation types in Europe during the Last Glacial Maximum using MaxEnt, based on modern data from the Siberian region, which has a climate similar to the European glacial climate, revealed the presence of areas with high and moderate climate suitability for temperate light-coniferous forests in Southern and Eastern Europe"?

L.55 use "rapidly" instead of "quickly".

L.55-59, do you mean here "Pinus sylvestris dominated forests" or, in general, Scots pine in all pure and mixed forests?

L.60-79 Revise the paragraph and present thoughts fluently and more coherently, and clearly state the importance/ connection of climate change to your studied forests.

Results.

L.96 The model of the potential range is mystical, with no annotation

Table 1, the used environmental/bioclimatic variables (use the uniform term for this) should have been explained beforehand in the Methods part, not in the Results. It is unclear how precipitation of the driest month was defined, what period is described, and the range for this variable and other predictor variables used in the modelling.

You should revise the names of environmental variables, e.g. Temperature seasonality-standard deviation of the monthly mean temperatures, which month, for what period? Season usually means a more extended period, months, but not one month. 

L.105, It is unclear how this Fig.1 was obtained since it does not describe the later application of the model. It should explain what the current potential range shows.

It is also unclear how this part of the world was selected to define the current potential range of pine-broadleaf forests. Isn't it so that those forests can be found elsewhere in Europe and Asia? Is it all Russia or part of it? 

You present levels of habitat suitability in legend. What units measure suitability, and how are these levels obtained (not explained anywhere in the manuscript)?

L.127 In Methods, it was reported 120 localities, not clear where from the 178 appeared. It would be needed an explanation of what those are.

L.131-133 It is unclear what you mean by this sentence. How these results on tree and herb layer structure was obtained? You do not describe any of it in Methods.

There are two Fig. 2!

In figures, habitat suitability could be marked in words, not numbers, for better readability.

L.184 Initially, only Southern Urals were mentioned, but now the area is extending. Was it part of the aims?

It seems that the sub-title for chapter 2.3.1. Change of potential range in the Southern Urals " could be removed.

Table 2. It is not clear why such a short -20-year -window was chosen in this analysis. How can such results be used/interpreted?

Discussion

L.285-287 Here and elsewhere in the Discussion, it is unclear how you established that pine-broadleaf forests are not widespread, e.g., on the Bugulma-Belebey Upland or Volga Upland. Have you conducted a field survey or used any other way to assess this? Where does this knowledge come from? Provide references.

L.288 How small are "small areas" here?

You use quite often (L.230, 237, 249, 347...) in the text term "modern", e.g. modern potential range, "modern broadleaf forests." what do you mean by this term? Specify for the reader what is modern in your study context.

Material and Methods

L.379 provide some references to software/developers.

L.381, can authors be more specific about what "data on 120 georeferenced localities" is included in the text? How was this data obtained (field sampling, databases, previous studies, what criteria were used for 120-point selection, what year ?) The study should be replicable, but the data information seems scarce. 

L.404 Similarly, more information regarding bioclimatic variables should be included (how many, which ones, and why the selected ones). Describe the modelling process better; it is unclear what kind of models this software produces and what dependent variables are. Provide annotation and a list of tested variables with discretions. Usually, when developing models, several are tested and compared with each other. Was this all done in your case?

L. 411 Abbreviations should be explained.

The language of the manuscript is quite good, but still, it would benefit from some language improvement (sentence structure, term use, grammar mistakes, etc.), especially in the Introduction part. 

Author Response

The current study focuses on the impact of climate change on pine-broadleaf ecotone in the Southern Ural region. It aims to define the current potential range of the Tilio-Pinenion sub-alliance and also to model the habitat suitability of studied forests under two different climate change scenarios. The research uses MaxEnt software and data on 120 georeferenced localities and some other data (climate, soil, elevation). 

Answer: Dear Reviewer. We are thank you for careful review of our article. Your recommendations and comments are very useful and improved the quality of manuscript. Below are the answers to your comments (in the text of article corrections are marked in blue color). 

The Methods part of this study description needs to be considerably supplemented. It is not entirely clear how some results were obtained here due to the deficiencies in this section.

Answer: We have added the necessary explanations in the Methods part.

The Discussion should be revised and needs to be considerably improved. Discussion is a crucial part of the research article, where authors should interpret the study's results, relate them to existing literature, and provide insights into the implications of their findings. I would expect to see the results of this study also compared/aligned with other studies to get to know how similar/different the outcome of this study is from other studies. Also, discuss how trustworthy your study results are and explain the implications of your findings. Why should the reader care about your study (theoretical/practical implications)? How might it be used in decision-making, practice, or policy? Provide suggestions for further research based on your findings and discuss limitations in the discussion part but not in the Conclusions. Avoid repeating the result section; interpret the findings and support your claims with evidence from the study and other literature. 

Answer: We have significantly revised the Discussion part according to your recommendations and added necessary explanations to the text. Also, the suggestions for further research based on the results and discussion about limitations of the method have been moved from the Conclusion part to the Discussion.

Consider improving the Conclusions part, where you should conclude the findings of your study.

 Answer: The conclusion section has been significantly revised and shortened.

Comments:

L.33, 43 use " in the transitional zone between..." instead of "in the zone of contact."

Answer: Thanks for the comment. Done.

  1. 40 " The role of pine (Pinus sylvestris L.) in the vegetation cover of the SU changed..."- Do you mean "share of pine"? Provide a reference for this statement.

Answer: We have provided citation for this statement.

L.44 "Sustainable ecotone mixed pine-broadleaved forests have appeared." what do you mean by "sustainable" here? In what way is it sustainable? Moreover, "have appeared"? Pay attention to grammar: shouldn't it be "ecotone of mixed pine-broadleaved forests"?

Answer: You're absolutely right. It is more correct to say not "sustainable ecotone mixed pine-broadleaf forests have appeared", but "ecotone mixed pine-broadleaf forests have been present for a long time".

In this phrase, "ecotone" is used as an adjective, not a noun.

L.47-51 Sentence is difficult to follow; consider improving. e.g. "In terms of floristic classification, these ecotone forests fall within the Carpino Fagetea sylvaticae class as designated by Jakucs ex Passarge in 1968. They are categorized under the order Carpinetalia betuli P.Fukarek 1968, within the alliance Aconito lycoctoni Tilion cordatae, which was described by Solomeshch et Grigoriev in Willner et al. in 2016. Furthermore, they are subcategorized under the suballiance Tilio cordatae Pinenion sylvestris, as proposed by Shirokikh et al. in 2021."

Answer: Thanks for the comment. We have corrected this sentence according to your recommendations.

L.51-55 unable to understand the sentence. Shouldn't it be: "Simulations of the distribution of vegetation types in Europe during the Last Glacial Maximum using MaxEnt, based on modern data from the Siberian region, which has a climate similar to the European glacial climate, revealed the presence of areas with high and moderate climate suitability for temperate light-coniferous forests in Southern and Eastern Europe"?

Answer: Thanks for the good wording. Corrected.

L.55 use "rapidly" instead of "quickly".

Answer: Done.

L.55-59, do you mean here "Pinus sylvestris dominated forests" or, in general, Scots pine in all pure and mixed forests?

Answer: The forests with Pinus sylvestris were rapidly replaced by forests with Picea abies and temperate deciduous forests 10,500 years ago. Necessary corrections have been made to the text of the manuscript.

L.60-79 Revise the paragraph and present thoughts fluently and more coherently, and clearly state the importance/ connection of climate change to your studied forests.

 Answer: We revised this paragraph.

Results.

L.96 The model of the potential range is mystical, with no annotation

Answer: Necessary corrections have been made to the text of the manuscript.

Table 1, the used environmental/bioclimatic variables (use the uniform term for this) should have been explained beforehand in the Methods part, not in the Results. It is unclear how precipitation of the driest month was defined, what period is described, and the range for this variable and other predictor variables used in the modelling.

Answer: Thanks for the comment. We moved the table of environmental variables from Results part to Methods part. We used CHELSA rasters of bioclimatic variables for modeling. They were calculated by the developers as averages for the last 30 years. They calculated for each pixel the driest month (with the least precipitation amount) or quarter, depending on the indicator. These periods could vary in different pixels of the raster and in different years. For example, in the temperate zone of the European part of the Russian Federation, the driest month is most often January, and the driest quarter is January through March, but this is not always the case. Since we cite references to primary sources, we believe that it would be superfluous to provide formulas for calculating these indicators in the methodology, since we use a ready product (rasters) without calculating these indicators ourselves.

You should revise the names of environmental variables, e.g. Temperature seasonality-standard deviation of the monthly mean temperatures, which month, for what period? Season usually means a more extended period, months, but not one month. 

Answer: Temperature Seasonality, Precipitation seasonality are terms used for environmental variables in the WorldClim (https://worldclim.com/) and CHELSA databases. These indices characterize the annual range of temperature and precipitation and are calculated as the standard deviation of monthly averages for 12 months. Since we cite references to primary sources, we believe that it would be superfluous to provide formulas for calculating these indicators in the methodology.

L.105, It is unclear how this Fig.1 was obtained since it does not describe the later application of the model. It should explain what the current potential range shows.

Answer: Necessary corrections have been made to the text of the manuscript.

It is also unclear how this part of the world was selected to define the current potential range of pine-broadleaf forests. Isn't it so that those forests can be found elsewhere in Europe and Asia? Is it all Russia or part of it? 

Answer: Raster layers covering the whole Eurasia were taken for modeling. The illustrations show only a part of the territory where suitable habitats of particular type of pine-broadleaf forests belonging to the suballiance Tilio-Pinenion are found. No suitable habitats of this particular forest type have been found in Asia and European countries. All necessary explanations are included in the text.

You present levels of habitat suitability in legend. What units measure suitability, and how are these levels obtained (not explained anywhere in the manuscript)?

Answer: The "cloglog" output gives a score between 0 (completely unsuitable) and 1 (completely consistent with the species or community optimum) of habitat suitability. The necessary explanations are included in the Methods part.

L.127 In Methods, it was reported 120 localities, not clear where from the 178 appeared. It would be needed an explanation of what those are.

Answer: It is recommended not to use geo-referenced points (=localities of plant communities) closer than 1 km to each other for modeling. Therefore, part of the geo-referenced points was removed and only 120 points were used in the modeling. All available 178 points of geobotanical descriptions were used to analyze the floristic composition of pine-broadleaf forests.

L.131-133 It is unclear what you mean by this sentence. How these results on tree and herb layer structure was obtained? You do not describe any of it in Methods.

Answer: Thanks for the comment. Based on the 178 complete relevés, a phytosociological spectrum of communities was compiled, intended to analyze changes of floristic composition of pine-broadleaf forests depending on habitat suitability. For this purpose, a buffer of 10X10 meter was created for each geo-referenced point (relevé locality). The polygons were superimposed on the model of potential range and habitat suitability in these localities was calculated using the "Zonal Statistics" module in QGIS. All herbaceous species mentioned in geobotanical relevés have been as-signed to vegetation classes for which this species is typical and where has an ecological optimum. After then, the proportion of each species in the projective cover of the herbaceous layer was calculated taking into account the constancy (percentage of occurrence in the group of relevés) and abundance of species (projective cover of species). The proportion of broad-leaved tree species in the understorey was also calculated in the same way. All necessary explanations are included in the Methods part.

There are two Fig. 2!

Answer: Done.

In figures, habitat suitability could be marked in words, not numbers, for better readability.

Answer: Done.

L.184 Initially, only Southern Urals were mentioned, but now the area is extending. Was it part of the aims?

Answer: For modeling we used 120 geo-referenced points (localities of geobotanical relevés) made by the authors of the manuscript during 2010-2023 within the Southern Ural region in the mountain-forest zone of the Bashkortostan Republic and on Ufa Plateau.  The modeling yielded a potential range that showed habitats with suitable conditions on the Volga Upland and in the Volga lowland region, as well as small areas with low suitability on the Bugulma-Belebey Upland and in the Middle Ural. According to literature sources, we established the presence of pine-broadleaf forests on the Volga Upland and in the Volga lowland region, which are similar to the pine-broadleaf forests of the Southern Ural, but differ from them in floristic composition.

It seems that the sub-title for chapter 2.3.1. Change of potential range in the Southern Urals " could be removed.

Answer: We changed the structure of the manuscript.

Table 2. It is not clear why such a short -20-year -window was chosen in this analysis. How can such results be used/interpreted?

 Answer: Forecasting models have been developed for the mid-21st century (2040s-2060s) and the second half of the 21st century (2060s-2080s). For convenience, many articles use the mean dates of 2050s and 2070s. For the avoidance of doubt, we have corrected the dates to match the primary sources. The period from 2020 to 2080. - is the deadline for which RCP4.5 and RCP8.5 modeling is foreseen. Further changes depend too much on the greenhouse gas mitigation measures that will or will not be taken, changes in land cover structure and other factors that may have synergistic effects.

Discussion

 L.285-287 Here and elsewhere in the Discussion, it is unclear how you established that pine-broadleaf forests are not widespread, e.g., on the Bugulma-Belebey Upland or Volga Upland. Have you conducted a field survey or used any other way to assess this? Where does this knowledge come from? Provide references.

Answer: Having thoroughly analyzed all available literature sources, we not found published geobotanical relevés of pine-broadleaf forests growing on the Bugulma-Belebey Upland. The authors of the article have evidences of the presence of pine-broadleaf forests on the Bugulma-Belebey Upland, but these communities differ from the forests belonging to the suballiance Tilio-Pinenion by floristic composition, and we did not use them for modeling. The pine-broadleaf forests growing in the habitats with similar ecological conditions on the Volga Upland and in the Volga lowland region were described in many literature sources, and these references are given in the manuscript. At the same time, the differences between Urals pine-broadleaf forests and pine-broadleaf forests of the Volga Upland and Volga lowland region increase with distance from the Southern Ural, and pine-broadleaf forests of the Volga Upland and Volga lowland belong to another alliance of ecologo-floristic classification.

L.288 How small are "small areas" here?

 Answer: On the Volga Upland, pine-broadleaf forests are fragmented and located in the lower parts of mountain slopes, in ravines and on steep riverbanks, without occupying significant areas. Apparently, the climatic conditions are too dry and hot for these communities. We have added a physical map to Figure 1 in order to clarify the current distribution area of these forests in question in the manuscript.

You use quite often (L.230, 237, 249, 347...) in the text term "modern", e.g. modern potential range, "modern broadleaf forests." what do you mean by this term? Specify for the reader what is modern in your study context.

 Answer: The current potential range reflects the distribution of habitats with suitable conditions for the growth of these forests at a given time. Necessary corrections have been made to the text of the manuscript.

Material and Methods

L.379 provide some references to software/developers.

Answer: Done.

L.381, can authors be more specific about what "data on 120 georeferenced localities" is included in the text? How was this data obtained (field sampling, databases, previous studies, what criteria were used for 120-point selection, what year?) The study should be replicable, but the data information seems scarce. 

Answer: For modeling, 120 geo-referenced geobotanical relevés  performed by the authors of the manuscript during 2010-2023 and classified according to the principles of vegetation classification according Brown-Blanquet approach were used. Necessary corrections have been made to the text of the manuscript.

L.404 Similarly, more information regarding bioclimatic variables should be included (how many, which ones, and why the selected ones). Describe the modelling process better; it is unclear what kind of models this software produces and what dependent variables are. Provide annotation and a list of tested variables with discretions. Usually, when developing models, several are tested and compared with each other. Was this all done in your case?

Answer: Thanks for the comment. We conducted modeling initially on the full set of environmental factors in order to find out their contribution to the potential range model. In parallel, we constructed a correlation matrix in order to exclude highly correlated ecological factors in the modeling. Based on the analysis of the contribution of ecological factors and the correlation matrix to the model, predictors for potential range modeling were selected. These same predictors were used in modeling changes in potential range under different climate change scenarios. We have added Table S1 "The estimates of relative contributions of the environmental variables to the MaxEnt model" and Table S2 "Correlation analysis of the environmental variables" to the Supplementary Materials to the manuscript.

  1. 411 Abbreviations should be explained.

Answer: Done.

Reviewer 3 Report

In general, the manuscript has improved. The manuscript requires some minor language corrections before it is suitable for publication.

Minor editing of English language required

Author Response

In general, the manuscript has improved. The manuscript requires some minor language corrections before it is suitable for publication.

Comments on the Quality of English Language

Minor editing of English language required 

Answer: Dear Reviewer. Thank you for review of our article. On your recommendation, we have made some edits to the English language of our manuscript.

Round 2

Reviewer 1 Report

I appreciate the extent to which the authors incorporated my suggestions and comments. In particular, the greater consistency of terms, added explanations of terms and methods, and a few key topic sentences. The addition of the general map panel in Figure 1, for example, serves as a crucial guide for understanding. The manuscript is much easier to follow, and the conclusions are much clearer. I also appreciate some added context, with a few more specific examples, in the Discussion.

I have a few additional, minor notes:

Line 88: "there" should be "their"

Line 95-96: "model changes in habitat suitability"

Line 112-113: This is a good example of a topic sentence that helped my understanding of the flow of the manuscript.

Line 148: "excerpt" should be "except"

Line 392-394: This is another good example of a helpful addition.

Table 1 in the Materials and Methods - should this be labelled Table 3?

I have made a few grammatical suggestions above.

Author Response

I appreciate the extent to which the authors incorporated my suggestions and comments. In particular, the greater consistency of terms, added explanations of terms and methods, and a few key topic sentences. The addition of the general map panel in Figure 1, for example, serves as a crucial guide for understanding. The manuscript is much easier to follow, and the conclusions are much clearer. I also appreciate some added context, with a few more specific examples, in the Discussion.

Answer: Dear reviewer. Thank you for your careful review of our revised article. Your recommendations have significantly improved the quality of the manuscript. Necessary corrections have been made in the text.

I have a few additional, minor notes:

Line 88: "there" should be "their".

Answer: Thank you! Done.

Line 95-96: "model changes in habitat suitability"

Answer: Corrections have been made.

Line 148: "excerpt" should be "except"

Answer: Thank you! Done.

Table 1 in the Materials and Methods - should this be labelled Table 3?

Answer: Thank you! Done.
